# Acute Coronary Syndrome with Non-Obstructive Plaque on Angiography and Features of Vulnerable Plaque on Intracoronary Optical Coherence Tomography

**DOI:** 10.3390/diagnostics13193118

**Published:** 2023-10-03

**Authors:** Clarissa Campo Dall’Orto, Rubens Pierry Ferreira Lopes, Lara Vilela Eurípedes, Gilvan Vilella Pinto Filho, Marcos Raphael da Silva

**Affiliations:** Department of Hemodynamic and Interventional Cardiology of the Advanced Hemodynamic Therapy Center, Brazilian Society of Health Support Hospital, Teixeira de Freitas 45987-088, Bahia, Brazil

**Keywords:** coronary atherosclerotic disease, acute coronary syndrome, vulnerable plaques, intravascular imaging, optical coherence tomography, percutaneous coronary intervention

## Abstract

Optical coherence tomography (OCT) has a high spatial resolution and is useful in identifying coronary lesions with high-risk features (vulnerable plaques). These plaques are strongly associated with acute coronary syndrome (ACS). In this report, we present the case of a 43-year-old male patient presenting with typical chest pain that began three hours prior to admission. The patient exhibited an elevation of the ST segments of the anterior and lateral walls. Invasive stratification revealed a 40% lesion in the middle segment of the left anterior descending (LAD) artery. The patient was given optimized clinical treatment as he had a nonobstructive lesion in the LAD at the time of angiography. During the treatment, the patient continued to complain of angina on exertion. A follow-up coronary angiography, along with OCT analysis of the middle-to-moderate lesion in the LAD, revealed a plaque predominantly rich in lipids with signs of vulnerability. A percutaneous coronary intervention was performed. The patient’s recovery was uneventful, and he was discharged the day after the procedure. This case illustrates the evolution of intravascular imaging, particularly OCT, in the detection of vulnerable plaques.

## 1. Introduction

Coronary artery disease (CAD) in general, and acute coronary syndrome (ACS) in particular, remains the leading causes of death worldwide, with enormous health implications globally [1]. They are also the leading causes of disability-adjusted life years lost worldwide [2]. There is a very broad spectrum of clinical presentation, with a range of myocardial ischemic states, including unstable angina, non-ST-segment elevation myocardial infarction (NSTEMI), or ST-segment elevation myocardial infarction (STEMI) [3].

Optical coherence tomography (OCT) can determine the possible mechanisms of ACS and guide decision-making for percutaneous coronary intervention (PCI). OCT is gaining popularity among interventional cardiologists, particularly in response to the increasing occurrence of complex coronary cases in interventional cardiology laboratories [4]. Although intravascular ultrasound (IVUS) is the most widely used intravascular imaging modality in recent years due to extensive supportive clinical data [5,6,7], OCT offers several distinct advantages. It provides a more accurate estimation of the calcified lesions, enhanced assessment of reasons for stent failure in cases of restenosis, and detailed information after PCI (tissue prolapse, dissection, thrombi, and presence of stent deposition), contributing to lower rates of major adverse cardiovascular events (MACEs) [8]. The high spatial resolution of OCT is useful in identifying coronary lesions with high-risk features, known as “vulnerable plaques”, which is crucial, as these plaques are associated with ACS. This is especially significant in patient populations at a high risk of future events, such as those with previous ACS or diabetes [9,10].

OCT can also help address several challenges regarding PCI in patients with ACS. Approximately 4–10% of patients with STEMI and over 30% of patients with NSTEMI have no identifiable culprit lesion on angiography. Additionally, more than 10% of patients with ACS have multiple culprit lesions on angiography [4]. In such cases, OCT helps in identifying these lesions, especially when angiography results are inconclusive.

We present an illustrative case of ACS involving nonobstructive plaques observed on angiography and vulnerable plaque characteristics observed through OCT.

## 2. Objective

The objective of this report was to emphasize the importance of identifying vulnerable plaques in high-risk patients, such as those with diabetes and previous ACS. Adequate management of these plaques can help to improve symptoms and possibly reduce cardiovascular events.

## 3. Case Report

The patient was a 43-year-old smoker with systemic arterial hypertension and non-insulin-dependent type 2 diabetes mellitus. He presented to the emergency room with chest pain that had started 3 h before admission. There were no previous similar episodes.

On admission, the patient exhibited hypertension with a blood pressure of 180/130 mmHg. The remaining physical examination was unremarkable, but the electrocardiogram displayed ST-segment elevation (Figure 1). He was orally administered 300 mg of aspirin and 180 mg of ticagrelor and transferred to the hemodynamics laboratory within a 60 min door-to-balloon time.

Cardiac catheterization was performed via the right radial artery (Figure 2). The procedure revealed a mild 30% lesion in the proximal segment of the right coronary artery (Figure 2A), circumflex artery with no obstructive lesions (Figure 2B) 40% lesion in the middle segment of the left anterior descending artery (LAD) (Figure 2C-D), and left ventriculography with mild hypokinesia (+/4+) on the anterior wall. During angiography, the LAD lesion was found to be non-obstructive with no evidence of thrombus, and with thrombolysis in myocardial infarction (TIMI) grade 3 flow, we opted to continue optimized clinical treatment.

Notable laboratory findings revealed that the peak troponin I level was 38.5 ng/mL; hematocrit 38.2%; hemoglobin 15.6 g/dL; peak leukocytes 16,000 cell/μL; platelets 250,000 platelets μ/L; fasting blood glucose 230 mg/dL; glycated hemoglobin 6.8%; low-density lipoprotein cholesterol (LDL) 149 mg/dL; high-density lipoprotein cholesterol (HDL) 42 mg/dL, and triglycerides 187 mg/dL. Other laboratory tests were normal.

Transthoracic Doppler echocardiography revealed mild hypokinesis in the anterior and apical of the left ventricle (LV), with an estimated LV function of 46%.

The optimized clinical treatment included oral administration of aspirin 100 mg once daily; ticagrelor 90 mg twice daily; metoprolol tartrate 50 mg twice daily; atorvastatin 80 mg once daily, and ramipril 2.5 mg/day. Prior to hospitalization, the patient was on metformin 1000 mg twice daily for controlling diabetes; however, during hospitalization, his blood glucose levels were poorly controlled. Hence, we suspended metformin and administered NPH insulin every 12 h (regular insulin was administered for prandial insulin control).

During the hospitalization, the patient continued to complain of angina on exertion for three days following his admission. Therefore, a follow-up coronary angiographic study was conducted (the findings were the same as in the previous coronary angiography, showing a mild-to-moderate obstruction in LAD), accompanied by intravascular imaging using OCT, to assess the mild-to-moderate lesion in LAD. OCT was not performed during the first coronary angiography due to the cost limitations imposed on hemodynamic services by health insurance in Brazil.

The OCT analysis of the lesion revealed plaques that were predominantly rich in lipids and signs of vulnerability. These signs included a lipid arch greater than 180°, thin-cap atheroma, plaque thickness of 70 μm, accumulation in macrophages, and minimum luminal area of 3.24 mm² (Figure 3A–C).

Therefore, we decided to perform a percutaneous coronary intervention (PCI) using two Resolute Onix stents (Medtronik, Minneapolis, MN, USA) of 3.0 × 26 mm^2^ and 3.0 × 40 mm^2^ with minimal overlapping of the struts in the proximal and middle segments of the LAD. Notably, no residual lesions or intracoronary complications were observed, and the TIMI flow was 3. OCT images post-PCI (Figure 3D–E) showed no dissection at the proximal and distal edges, with a well-apposed stent along its entire length demonstrating excellent expansion, with the point of least expansion at 105% of the desired expansion. The proximal and distal reference segment lumens (≥5 mm from the edge of the stent) each had a minimum luminal area (MLA) of ≥4.5 mm^2^.

After the procedure, the patient’s recovery progressed without complications. He was asymptomatic and was discharged the next day. The treatment plan included dual antiplatelet therapy with acetylsalicylic acid (100 mg/day) and ticagrelor (90 mg twice a day), in addition to metoprolol succinate 50 mg twice daily, atorvastatin 80 mg once daily, ramipril 10 mg/day, metformin 1000 mg twice daily (which was reintroduced before discharge), and dapagliflozin 10 mg once daily.

During the 6-month follow-up, the patient remained asymptomatic and underwent a control ischemic test using cardiac magnetic resonance, which revealed a small area of fibrosis on the anterior wall and no signs of ischemia.

## 4. Discussion

The most common mechanisms underlying ACS are plaque rupture and erosion, and less frequently involved conditions include thrombosis triggered by calcified nodules, spontaneous dissection, and stent thrombosis. Owing to its higher resolution, OCT has a clear advantage over IVUS in identifying the pathological mechanisms of ACS [9].

Vulnerable plaques were first described by Muller et al. in the 1980s [11]. Clinically, vulnerable plaques are atherosclerotic lesions that have the potential to progress to thrombosis, thereby increasing the risk of MACE. Pathologically, these plaques consist of a large lipid core rich in inflammatory cells, such as macrophages, and a thin fibrous cap. These plaques are often associated with microcalcifications, areas of hemorrhage, and neovascularization [12,13,14,15,16].

Several studies have used intravascular imaging techniques to identify clinical and injury-related factors (vulnerability characteristics) that put patients with ACS at greater risk of MACE [13,17]. Among these risk factors, diabetes mellitus was the strongest predictor of adverse events. Lesion-related adverse cardiovascular events were more frequently observed in thin-cap fibroatheromas (fibrous cap < 75 mμ), with a large plaque burden (plaque burden > 70%), small AML (<3.5–4 mm^2^), lipid range > 180°, and presence of macrophages (the latter was detected by OCT) [18].

The COMBINE OCT-FFR study included patients with diabetes and an angiographically intermediate lesion, stable CAD or ACS (in this case, for the evaluation of non-culprit lesions), and negative invasive coronary physiology for ischemia. OCT evaluations revealed that patients with thin cap fibroatheroma (TCFA) had a higher MACE rate and that TCFA was present in 25% of the patients with diabetes. Further, OCT-identified TCFA was associated with up to a fivefold higher risk of MACE, even in the absence of ischemia [17]. In our case, an evaluation with invasive physiology was not considered because it is contraindicated for blood vessels responsible for acute infarction (culprit lesions) due to microcirculation dysfunction and plaque instability. Moreover, several studies have shown that the analysis of physiology invasive therapy is clinically ineffective in this setting [19,20,21,22].

In the context of ACS, recent research has shown that OCT allows for the diagnostic investigation of myocardial infarction in coronary arteries with non-obstructive lesions (MINOCA), defined as coronary artery disease with <50% diameter stenosis on angiography [23]. MINOCA can present as either STEMI or NSTEMI, which was a diagnosis initially suggested in our case. It should be noted that despite being more prevalent in females, MINOCA can also affect males.

We attributed the cause of our patient’s ACS to plaque erosion, as endothelial denudation was observed on OCT with consequent probable distal embolization. Plaque erosion accounts for 25% of ACS cases and results from disruption of the endothelial layer covering the plaque. Once the thrombus is treated, plaque thrombogenicity is limited to the exposed subendothelial surface and can be treated with antithrombotic therapy alone without the need for stent implantation [24,25,26,27]. However, our patient remained symptomatic, and we opted for a mechanical passivation of the plaque with stent in addition to the established pharmacology.

Exertional angina was reported in the days following the index event, despite coronary angiography showing only mild-to-moderate obstruction of the LAD that was confirmed in the second angiograph. These persisting symptoms may be due to the smaller size of the MLA (3.24 mm^2^ by OCT) in the proximal LAD. Angiographically mild-moderate lesions often cause functional repercussions, as evidenced by studies on invasive physiology in stable CAD [28]. Therefore, we opted for intravascular imaging instead of invasive physiology analysis.

The prominent advantages of OCT in patients with ACS include identification of the culprit lesion, characterization of the underlying pathology, and stent optimization [29]. As in our case, OCT assessment allows for the identification of plaque characteristics that are associated with high-risk factors, including diabetes and ACS, and helps clinicians determine whether aggressive treatment is beneficial for the patient.

A limitation of our case is that OCT was not performed during the first coronary angiography, due to the cost limitations imposed on hemodynamic services by health insurance in Brazil. Another limitation is that cardiac magnetic resonance (CMR) was not performed during hospitalization. When used early in the diagnosis of MINOCA, CMR can exclude myocarditis, takotsubo and other cardiomyopathies, as well as provide imaging confirmation of myocardial infarction [30,31]. However, CMR does not always uncover the cause of MINOCA, as in cases of absent or unavailable subendocardial or transmural ischemic late-gadolinium enhancement patterns, and will thus require further intravascular imaging [32]. Therefore, multimodality imaging with coronary OCT and CMR has recently been recommended for the diagnosis of MINOCA [23]. Unfortunately, neither CMR nor OCT are widely available, and in specialized centers diagnostic routes will vary based on available technology. Further research is needed to expand options for the challenging and complex diagnosis of the underlying causes of MINOCA.

## 5. Conclusions

In conclusion, we propose that in addition to the well-documented advantage of intravascular imaging methods over angiography in predicting the severity of lesions in ACS, OCT is also a valuable tool for the identification of the culprit lesion, characterization of the underlying pathology, and stent optimization.

## Figures and Tables

**Figure 1 diagnostics-13-03118-f001:**
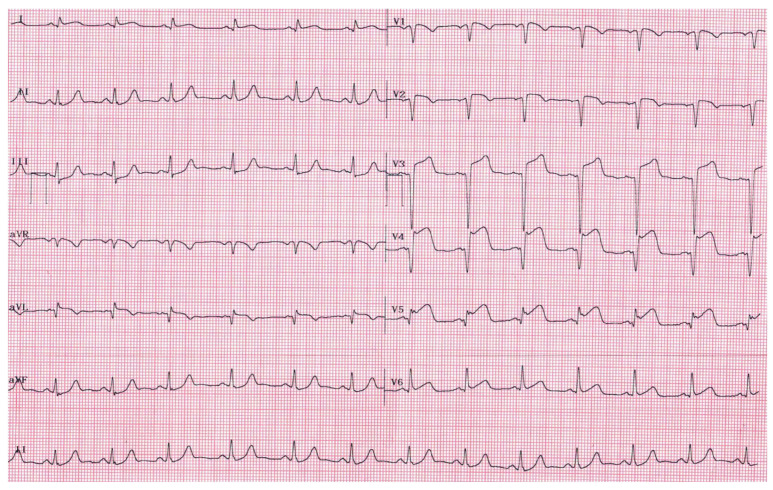
Electrocardiogram showing ST-segment elevation in the anterior (V1–V4) and lateral leads (V5, V6, I, aVL) with concomitant Q waves in V1–V4, I, and aVL.

**Figure 2 diagnostics-13-03118-f002:**
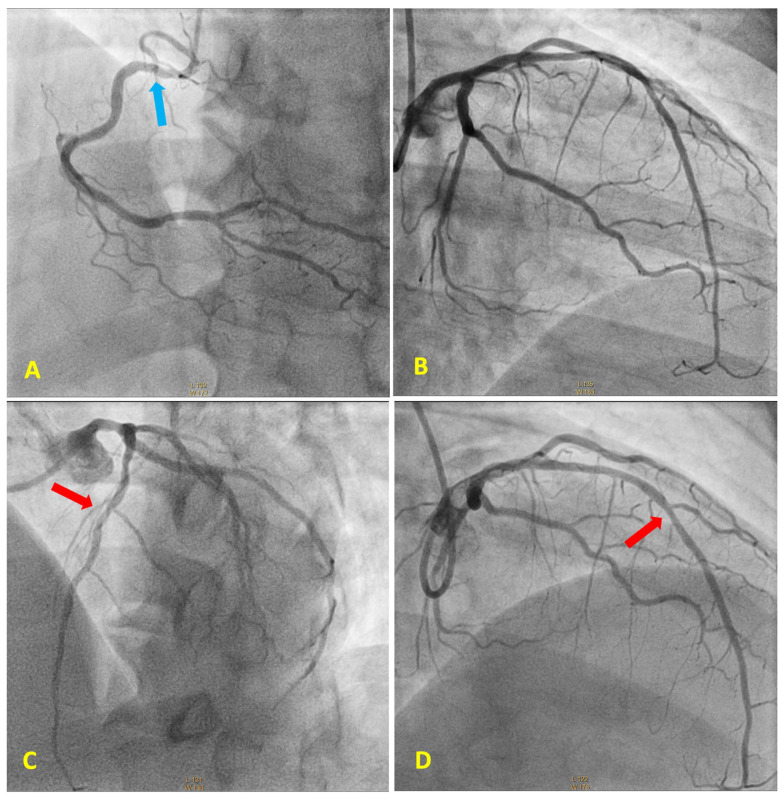
Coronary angiography. (**A**) Right coronary artery with 30% mild lesion in the proximal segment, blue arrow; (**B**) circumflex artery with no obstructive lesions; (**C**,**D**) left anterior descending (LAD) artery with 40% lesion in the middle segment, red arrows, with no evidence of thrombus and TIMI 3 flow. TIMI—Thrombolysis in myocardial infarction.

**Figure 3 diagnostics-13-03118-f003:**
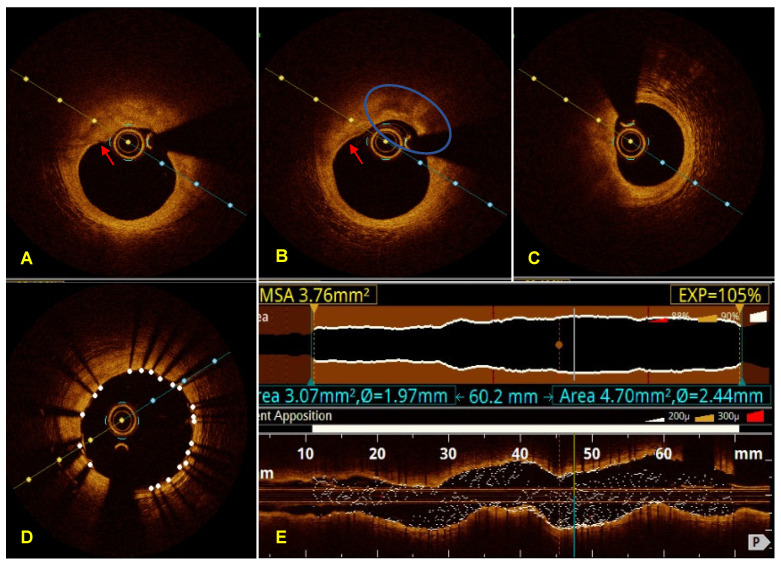
Optical coherence tomography of LAD. Cross-sectional images pre-PCI: (**A**) thin cap atheroma with an irregular luminal surface, suggestive of endothelial denudation (red arrow); (**B**) thin cap atheroma (red arrow) with abundant macrophages (blue circle); (**C**) plaque with a lipid arch greater than 180°; (**D**) cross-sectional image post-PCI at the smallest point displays an MLA of 7.41 mm^2^; (**E**) OCT post-PCI with lumen profile and rendered stent on the longitudinal OCT image using the tapered reference mode, with the point of smallest expansion being 105% of the desired expansion (in yellow). The OCT lumen profile shows the required total stent length of 60.2 mm (in blue). LAD—left anterior descending; PCI—percutaneous coronary intervention; MLA—minimum lumen area; OCT—optical coherence tomography.

## Data Availability

All data are reported in the text.

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
