# Peer review of "Acute Coronary Syndrome with Non-Obstructive Plaque on Angiography and Features of Vulnerable Plaque on Intracoronary Optical Coherence Tomography"

_diagnostics, 2023, doi:10.3390/diagnostics13193118_

Round 1

Reviewer 1 Report

The present is a very nice case report. Some issues need to be addressed.

1) although the ecg is not diagnostic, authors should better explain why they did not perform OCT in the first angio

2) in this case choice between imaging and functional evaluation may not be immediate (quote on PMID: 28629622)

3) why was the patient discharged in clopidogrel? and not prasugrel?

Good

Reviewer 2 Report

The clinical case presented by Clarissa Campo Dall'Orto et al. highlighted the role of OCT in detecting high-risk coronary plaque. 

The clinical case presented lacks clarity, as essential data such as troponin, laboratory and echocardiographic findings are missing. Furthermore, FFR and iFR were not tested during the index procedure. Finally, a CMR was not performed, although it is recommended in this context (MINOCA).

Moderate English revision is recommended. 

Reviewer 3 Report

Dear Editor,

I read with interest the case report by Clarissa Campo Dall'Orto et al. regarding the importance of identifying vulnerable plaques in high-risk patients, such as those with diabetes and previous ACS.

The conclusion of this study was that the current diagnostic methods using intravascular imaging, particularly OCT, provide 167 patients and their attending physicians with invaluable insights into the pathophysiology 168 of plaques. This is especially relevant in ACS cases that require aggressive treatments such 169 as drug therapy, which plays a crucial role in atherosclerotic plaque stabilization, as well 170 as various revascularization methods.

The manuscript is well written and easy to follow. The history of the patients is very well described, as well as the very suggestive paraclinical examinations. The paper is a clean, very easy to follow with important clinical impact. The manuscript can be accepted for publication in its current form. I want to congratulate the authors for their work.

Author Response

Thank you very much for appreciating our study. They were very important to us.

Reviewer 4 Report

Thank you for the opportunity to review this interesting case report. Indeed, optical coherence tomography (OCT) has gained popularity in the cath labs, particularly in assessing intermediate coronary lesions in the setting of acute coronary syndrome (ACS). In this report, a 43-year-old male patient with typical chest pain and ST-segment elevation in the anterior wall underwent coronary angiography which revealed a 40% lesion in the middle segment of LAD artery. Initially, the patient was kept on optimized clinical treatment but due to recurrence of chest pain he had a repeat coronary angiography with OCT analysis. The OCT images revealed a plaque with signs of vulnerability, and then a percutaneous coronary intervention was performed and afterwards the patient remained asymptomatic.

I would like to share the following considerations

1.
 The description of ECG in Figure 1 is incomplete. There is ST-segment elevation not only in anterior leads but also in lateral leads (V5, V6, I, aVL) with concomitant Q waves in V1-V4, I and aVL. 

2) Line 73: what do you mean “invasive stratification”. Make it simpler. “Cardiac catheterization was performed”

3) Please put arrows to indicate the stenoses in Figure 2.

4) line 90: “lipid morphology” is rather inappropriate term. Instead of “predominantly lipid morphology” please write “predominantly rich in lipids”

5) there is no information on glucose and lipids levels on admission and on cardiac troponin levels (extension of myocardial necrosis)

6) It is not clear to me the antithrombotic treatment given after the first coronary angiography

7) why clopidogrel was  preferred instead of ticagrelor which is the antiplatelet treatment of choice on top of aspirin

8) The major question which requires answer is the cause of patient’s effort angina. Effort angina suggests the presence of significant stenosis (>70%). What was the LAD anatomy on the repeat coronary angiography? Was the cause of effort angina thrombotic stenosis?

9) In conclusions, the message of this case report regarding the place of OCT is unclear

Only minor suggestions in my comments

Round 2

Reviewer 2 Report

I sincerely commend the Authors for their dedicated efforts in enhancing the clinical case. However, it is essential to acknowledge that there are still significant issues that require attention.

1.     Could you kindly provide the URL and essay detailing your hs-troponin data? Is the troponin level of 38.5 ng/L truly indicative of the peak troponin value?

2.     The inclusion of a CMR evaluation in suspected MINOCA cases is now of paramount importance. I really appreciate the incorporation of a CMR assessment during the follow-up, revealing a small ischemic scar in the LAD territory. 

a) It would be highly beneficial to incorporate an illustrative figure of the LGE sequence. 

b) Additionally, I recommend an expansion of the introduction and discussion sections, particularly emphasizing the pivotal role of CMR in cases of MINOCA. It is worth noting that the evidence supporting early CMR in all MINOCA has evolved significantly. In fact, recent studies (e.g. DOI 10.1016/j.jcmg.2022.12.029; DOI https://doi.org/10.1016/j.jcmg.2023.05.016), have underscored the relevant diagnostic and prognostic significance of early CMR in ischemic MINOCA. In addition, a comprehensive echocardiographic evaluation, including strain analyses, could be useful to detect subtle CAD, as in your case (e.g. 10.1093/ehjci/jead046;https://doi.org/10.1161/CIRCIMAGING.110.959817). This comprehensive approach could significantly enhance the overall depth and relevance of the clinical case.

3. An English revision to enhance clarity and fluency is recommended.

Moderate English revision is necessary. I recommend double-checking abbreviations and typos.

Author Response

I sincerely commend the Authors for their dedicated efforts in enhancing the clinical case. However, it is essential to acknowledge that there are still significant issues that require attention.

  1. Could you kindly provide the URL and essay detailing your hs-troponin data? Is the troponin level of 38.5 ng/L truly indicative of the peak troponin value?

Thank you for your suggestion. We have added the URL to the text at line 88. Please note that the manuscript does report the correct, plausible value of 38.5 ng/mL.

  1. The inclusion of a CMR evaluation in suspected MINOCA cases is now of paramount importance. I really appreciate the incorporation of a CMR assessment during the follow-up, revealing a small ischemic scar in the LAD territory. 
  2. a) It would be highly beneficial to incorporate an illustrative figure of the LGE sequence. 
  3. b) Additionally, I recommend an expansion of the introduction and discussion sections, particularly emphasizing the pivotal role of CMR in cases of MINOCA. It is worth noting that the evidence supporting early CMR in all MINOCA has evolved significantly. In fact, recent studies (e.g. DOI 10.1016/j.jcmg.2022.12.029; DOI https://doi.org/10.1016/j.jcmg.2023.05.016), have underscored the relevant diagnostic and prognostic significance of early CMR in ischemic MINOCA. In addition, a comprehensive echocardiographic evaluation, including strain analyses, could be useful to detect subtle CAD, as in your case (g. 10.1093/ehjci/jead046; https://doi.org/10.1161/CIRCIMAGING.

110.959817). This comprehensive approach could significantly enhance the overall depth and relevance of the clinical case.

Thank you for your interest in this topic. We strongly agree with the importance of CMR imaging in the diagnosis of MINOCA, but are focusing our article on the use of optical coherence tomography (OCT) in the diagnosis of vulnerable plaques. The topic is relevant to patients with MINOCA, but also to patients with greater than 50% stenosis and meeting plaque vulnerability criteria. These patients would not be classified as having MINOCA.

In our patient’s case, we had already diagnosed MINOCA in the hospital phase using coronary angiography. We had also identified anatomical high-risk markers (vulnerable plaques) on OCT in addition to clinical high-risk criteria (e.g., recent acute coronary syndrome, presence of diabetes). We believe that CMR in this patient's in-hospital phase was thus not necessary as we used other resources to arrive at the diagnosis and properly manage the case. It was used, however, at the patient’s follow-up visit and our active investigation of post-revascularization ischemia.

CMR is clearly important to the accurate diagnosis of MINOCA if the cause of the MI is not identified in the catheterization laboratory (which it had been in our case). CMR may also be helpful in confirming a diagnosis if the history or echocardiogram suggest takotsubo cardiomyopathy or myocarditis. The Multisociety Consensus Quality Improvement Revised Consensus Statement for Endovascular Therapy of Acute Ischemic Stroke (doi.org/10.1016/j.jcmg.2023.05.016) has also noted that: “in a meta-analysis of 26 studies with more than 3,600 patients, a specific cause of MINOCA was identified in only 22% of patients. A subsequent CMR helped reclassify 68 percent of these patients to a different MINOCA etiology.”

In short, although we could have performed CMR during the patient’s hospitalization, it would not have provided information that would have changed the patient’s risk classification or the conduct of the case. In particular, left ventriculography and echocardiogram showed segmental deficit in the anterior wall, while coronary angiography showed signs of plaque vulnerability pointing to erosion as a causal factor. As the emphasis of this report is the use of OCT to identify vulnerable plaque, we feel that extending manuscript text too far out of focus unnecessarily adds to the word count and reader fatigue.

Regarding the echocardiogram, it was performed at the patient’s bedside and the echocardiographer in our hospital does not perform strain analyses in such situations. We recognize the importance of this test for detecting subtle CAD, but in our case it had already been detected.

  1. An English revision to enhance clarity and fluency is recommended.

Thank you for noting this. We have had our manuscript re-edited by our original English editing service (Editage); the proofreading certificate is attached. However, if the editors feel that the language is still not adequate, we are willing to send the manuscript to another company.

Reviewer 4 Report

The authors have satisfactorily dealt with my comments

Author Response

Thank you for your considerations.

Round 3

Reviewer 2 Report

I want to express my sincere appreciation to the authors for their diligent work in revising this case report. 

However, I would like to point out that there is a crucial aspect that needs further elaboration: the absence of a CMR examination during the patient's hospitalization in the context of MINOCA.

As previously mentioned in my revision, CMR plays a pivotal role in differentiating MINOCA mimickers such as myocarditis, as demonstrated in various studies (e.g. 10.1016/j.jcmg.2022.12.029; 10.1093/ehjci/jead182; 10.1016/j.jcmg.2021.02.021; https://doi.org/10.1016/j.jcmg.2020.05.045). 

Additionally, CMR can provide invaluable information for precise risk stratification (e.g. 10.1016/j.jcmg.2023.05.016; 10.1161/CIRCIMAGING.122.014454; https://doi.org/10.3390/jcm12062266)

It's important to note that the cost of a CMR is considerably lower than that of a subsequent coronary angiography with OCT, as detailed in this case report. Furthermore, had a comprehensive CMR been performed following the initial coronary angiography, it could have significantly influenced subsequent management decisions. For instance, CMR can help determine critical factors such as the presence of ischemic LGE, the presence of edema, and an accurate evaluation of cardiac function.

I respectfully disagree with the authors' portrayal of CMR's utility in this context, which we recognize as being essential. Therefore, I strongly suggest the inclusion of this aspect in the discussion section of the case report, even if it is acknowledged as a limitation, which we know could be common in daily clinical practice. This addition would enhance the comprehensiveness and clinical relevance of the report.

I have not found the proofreading certificate. However, some errors are still present, such as 'plaqueses' as a keyword.

Author Response

We thank Reviewer 2 for continued support of our manuscript. We agree that it is important to note that the cost of CMR is considerably less than that of a subsequent coronary angiography with OCT. However, we respectfully disagree that a comprehensive CMR after the initial coronary angiography would have significantly influenced subsequent management decisions for our patient. After the first coronary angiography showed the mild-moderate lesion in the coronary LAD, if CMR revealed ischemia in the anterior wall the patient would have to return to the hemodynamics room so that the lesion could be approached with angioplasty with stent. This would not contraindicate the use of OCT as an adjunct intra-vascular imaging method (IVUS could have been used as well). Key advantages of OCT in patients with ACS include identification of the culprit lesion, characterization of underlying pathology, and stent optimization [1-3]. The value of intra-vascular imaging in stent optimization for patients with complex coronary artery lesions was recently demonstrated by Lee et al. (2023), who found that mortality from cardiac causes, target vessel-related myocardium, or clinically-guided target vessel revascularization was lower for intravascular image-guided PCI than for angiography-guided PCI [4].

[1] Ali, Z.A.; Karimi Galougahi, K.; Mintz, G.S.; Maehara, A.; Shlofmitz, R.A.; Mattesini, A. Intracoronary optical coherence tomography: state of the art and future directions. EuroIntervention 2021, 17, e105–e123. DOI:10.4244/EIJ-D-21-00089, PMID: 34110288. PMCID: PMC9725016.

[2] Johnson, T.W.; Räber, L.; Di Mario, C.; Bourantas, C.V.; Jia, H.; Mattesini, A.; Gonzalo, N.; de la Torre Hernandez, J.M.; Prati, F.; Koskinas, K.C.; et al. Clinical use of intracoronary imaging. Part 2: Acute coronary syndromes, ambiguous coronary angiography findings, and guiding interventional decision-making: an expert consensus document of the European Association of Percutaneous Cardiovascular Interventions. EuroIntervention 2019, 15, 434–451. DOI:10.4244/EIJY19M06_02.

[3] Gupta, A.; Shrivastava, A.; Vijayvergiya, R.; Chhikara, S.; Datta, R.; Aziz, A.; Singh Meena, D.; Nath, R.K.; Kumar, J.R. Optical coherence tomography: an eye into the coronary artery. Front Cardiovasc Med 2022, 9, 854554. DOI:10.3389/fcvm.2022.854554, PMID: 35647059. PMCID: PMC9130606.

[4] Lee JM, Choi KH, Song YB, Lee JY, Lee SJ, Lee SY, et al; RENOVATE-COMPLEX-PCI Investigators. Intravascular Imaging-Guided or Angiography-Guided Complex PCI. N Engl J Med. 2023 May 4;388(18):1668-1679. doi: 10.1056/NEJMoa2216607. Epub 2023 Mar 5. PMID: 36876735.

We also respectfully disagree with the Reviewer’s position on the usefulness of CMR in the context of our case report, given the complementary tests (echocardiogram, coronary angiography, optical coherence tomography) that were performed. Nonetheless, we have inserted the following as a study limitation, lines 196-206:

A limitation of our case is that cardiac magnetic resonance (CMR) was not performed during hospitalization. When used early in the diagnosis of MINOCA, CMR can exclude myocarditis, takotsubo and other cardiomyopathies, as well as provide imaging confirmation of AMI [30,31]. However, CMR does not always uncover the cause of MINOCA, as in cases of absent or unavailable subendocardial or transmural ischemic late-gadolinium enhancement patterns, and will thus require further intravascular imaging [32]. Therefore, multimodality imaging with coronary OCT and CMR has recently been recommended for the diagnosis of MINOCA [33]. Unfortunately, neither CMR nor OCT are widely available, and in specialized centers diagnostic routes will vary based on available technology. Further research is needed to expand options for the challenging and complex diagnosis of underlying causes of MINOCA.

We appreciate the three bibliographic references suggested by the Reviewer and have added them to the text and the reference list:

  1. Sörensson, P.; Ekenbäck, C.; Lundin, M.; et al. Early comprehensive cardiovascular magnetic resonance imaging in patients with myocardial infarction with nonobstructive coronary arteries. JACC Cardiovasc Imaging. 2021, 14, 1774­1783. DOI: 10.1016/j.jcmg.2021.02.021. Epub 2021 Apr 14. PMID: 33865778.
  2. Mileva, N.; Paolisso, P.; Gallinoro, E.; et al. Diagnostic and prognostic role of cardiac magnetic resonance in MINOCA: systematic review and meta-analysis. JACC Cardiovasc Imaging. 2023, 16, 376­389. DOI: 10.1016/j.jcmg.2022.12.029. PMID: 36889851.
  3. Reynolds, H.R.; Maehara, A.; Kwong, R.Y.; et al. Coronary optical coherence tomography and cardiac magnetic resonance imaging to determine underlying causes of myocardial infarction with nonobstructive coronary arteries in women. Circulation. 2021, 143, 624­640. DOI: 10.1161/CIRCULATIONAHA.120.052008.

All stages of writing and reviewing the manuscript were reviewed. We have 3 revision certificates, but to send a file here I can only send one, I chose the most recent revision certificate.
